# Approximate Bayesian Inference for a Mechanistic Model of Vesicle Release at a Ribbon Synapse

**Cornelius Schröder**[*]
Institute for Ophthalmic Research
University of Tübingen
cornelius.schroeder@uni-tuebingen.de

**Ben James**[*]
School of Life Sciences
University of Sussex
bmjame02@gmail.com

**Leon Lagnado**
School of Life Sciences
University of Sussex
l.lagnado@sussex.ac.uk

**Philipp Berens**
Institute for Ophthalmic Research
University of Tübingen
philipp.berens@uni-tuebingen.de

## Abstract

The inherent noise of neural systems makes it difficult to construct models which accurately capture experimental measurements of their activity. While much research has been done on how to efficiently model neural activity with descriptive models such as linear-nonlinear-models (LN), Bayesian inference for mechanistic models has received considerably less attention. One reason for this is that these models typically lead to intractable likelihoods and thus make parameter inference difficult. Here, we develop an approximate Bayesian inference scheme for a fully stochastic, biophysically inspired model of glutamate release at the ribbon synapse, a highly specialized synapse found in different sensory systems. The model translates known structural features of the ribbon synapse into a set of stochastically coupled equations. We approximate the posterior distributions by updating a parametric prior distribution via Bayesian updating rules and show that model parameters can be efficiently estimated for synthetic and experimental data from in vivo two-photon experiments in the zebrafish retina. Also, we find that the model captures complex properties of the synaptic release such as the temporal precision and outperforms a standard GLM. Our framework provides a viable path forward for linking mechanistic models of neural activity to measured data.

## 1 Introduction

The activity of sensory neurons is noisy — a central goal of systems neuroscience has therefore been to devise probabilistic models that allow to model the stimulus-response relationship of such neurons while capturing their variability [1]. Specifically, linear-nonlinear (LN) models and their generalizations have been used extensively to describe neural activity in the retina [2, 3]. However, these type of models cannot yield insights into the mechanistic foundations of the neural computations they aim to describe, as they do not model their biophysical basis. On the other hand, mechanistic models on the cellular or subcellular level have been rarely used to model stimulus-response relationships: they require highly specialized experiments to estimate individual parameters [4, 5], making it difficult to employ them directly in a stimulus-response model; alternatively, they often result in an intractable likelihood, making parameter inference challenging [6].

---

[*]Equal contribution. Code available at https://github.com/berenslab/abc-ribbon

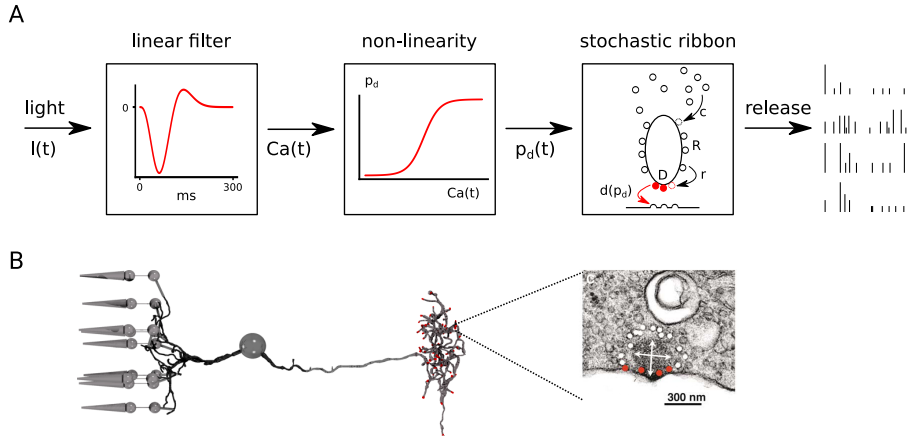

Figure 1: **Overview of the model. A.** After a linear-non-linear processing stage, the signal is passed to a biophysically inspired model of a ribbon synapse in which vesicles are released in discrete events. **B.** Sketch of a bipolar cell with attached photoreceptors (left) and a high resolution electron microscopy (EM) image of a ribbon synapse with its vesicle pools. The readily releasable pool is highlighted in red, the reserve pool is shown in white (EM image adapted from [14]).

Here we make use of recent advances in approximate Bayesian computation (ABC) [6, 7, 8, 9, 10, 11] to fit a fully stochastic, biophysically inspired model of vesicle release from the bipolar cell (BC) axon terminal to functional two-photon imaging data from the zebrafish retina (Fig. 1). It includes a linear-nonlinear stage to model the stimulus dependency, and a set of stochastically coupled equations modeling biophysical properties of the BC synapse. At this so-called "ribbon synapse", a specialized protein complex, the "ribbon", acts as a conveyor belt that "tethers" and "loads" vesicles onto active zones for future release [12, 13]. It organizes vesicles into multiple pools: the "docked" (or readily releasable) pool consists of a number of vesicles located directly above the plasma membrane, while the "ribboned" pool consists of vesicles attached to the ribbon further from the cell membrane. The docked vesicles are thus primed for immediate release and can be released simultaneously (so called multivesicular release, MVR). The ribboned vesicles are held in reserve to refill the docked pool as it is depleted by exocytosis [14, 15]. The transitions of vesicles between those pools can be modeled by a set of coupled differential equations [16, 4], which we extend to a stochastic treatment. In addition to photoreceptors and bipolar cells in the retina [17], ribbon synapses are featured in many other sensory systems, such as in auditory hair cells and the vestibular system [18].

Thus, our proposed Bayesian framework links stimulus-response modeling to a biophysically inspired, mechanistic model of the ribbon synapse. This may contribute to a better understanding of sensory computations across levels of description, with applications in a diverse range of sensory systems.

## 2 Previous work

**Models of neural activity**    Variants and extensions of LN models have been widely used to model the activity of retinal neurons [2, 19, 1, 3]. In these descriptive models, the excitatory drive to a cell is modeled as the convolution of a receptive field kernel with the stimulus, followed by a static nonlinearity. The result of this computation sets the rate of a stochastic spike generator, most commonly using either a Binomial or Poisson distribution. These basic models have also been used to approximate BC activity [20], however they do not explicitly model the dynamics of vesicle release at the ribbon synapse. Existing mechanistic models of synaptic release often require highly specialized experiments to estimate parameters [21] or make only indirect inferences based on the spiking activity of post-synaptic cells [22, 23]. In addition, they have not been used to perform system identification. The linear-nonlinear kinetics (LNK) model [24] attempts to address this issue. After an initial LN stage, the LNK model passes this information into a "kinetics block" consisting of a first-order set of kinetic equations implicitly representing the availability of vesicles. However, the LNK model treats the states of the pools as rescaled Markov process and cannot easily account for discrete vesicle release or MVR at the given noise level of single synapses.

Table 1: Variables, parameters and distributions of the model.

| Variable | Description | Parameter | Movement distribution | Distribution |
|---|---|---|---|---|
| | time stretch of the kernel | $\gamma$ | | $\mathcal{N}(\mu, \sigma^2)$ |
| | non-linearity | $k, h$ | | $\mathcal{N}(\mu, \Sigma)$ |
| | correlation of exocytosed vesicles | $\rho$ | | $\mathcal{N}(\mu, \sigma^2)$ |
| $d_t$ | exocytosed vesicles | $p_{d_t}$ | Beta-Bin | |
| $D$ | vesicles at dock | | | |
| $r$ | vesicles ribbon $\rightarrow$ dock | $p_r$ | res. Binomial | $\mathcal{N}(\mu, \sigma^2)$ |
| $R$ | vesicles on ribbon | | | |
| $c$ | vesicles cytoplasm $\rightarrow$ ribbon | $\lambda_c$ | res. Poisson | $\Gamma$ |

We address these issues by proposing a model that combines LN modeling for system identification with a probabilistic, biophysically inspired model for the ribbon synapse, with the capability to model discrete, multi-vesicular release. In contrast to classical LN models, the parameters of this model are readily interpretable as they directly refer to biological processes.

**Approximate Bayesian Computation**   Many mechanistic models in computational neuroscience only provide means to simulate data and do not yield an explicit likelihood function. Therefore, their parameters cannot be inferred easily. In such simulator-based models, Bayesian inference can be performed through techniques known as Approximate Bayesian Computation or likelihood-free inference [8]. The general inference problem can be defined as follows: given some experimental data $\mathbf{x}_0$ and a mechanistic, simulator-based model $p(\mathbf{x}|\theta)$ parametrized by $\theta$, we want to approximate the posterior distribution $p(\theta|\mathbf{x} = \mathbf{x}_0)$. The simulator model allows us to generate samples $\mathbf{x}_i$ given any parameter $\theta$, but the likelihood function cannot be evaluated. Often, $\mathbf{x}_i$ is first mapped to a low dimensional space (so called "summary statistics"), in which a loss function is computed. This mapping defines the features the model is trying to capture [10].

There are two main approaches to solve the inference problem: (1) approximate the likelihood $p(\mathbf{x_0}|\theta)$ and then sample (e.g. via MCMC) to get the posterior [8, 10]. In this approach, guided sampling is often used to generate new samples and either train a neural network or update other parametric models for the likelihood [8, 9]. One disadvantage of this approach is that there is a second sampling step necessary to obtain the posterior, which can be as time consuming as the inference of the likelihood. (2) approximate the posterior $p(\theta|\mathbf{x} = \mathbf{x}_0)$. In principle, inference via rejection sampling could be applied, but is often inefficient. Thus, recently proposed methods use parametric models (like a mixture of Gaussian) to approximate the posterior over several sampling rounds [6]. In our work, we use an ABC method of type (2) with parametric prior distributions and Bayesian updating rules to approximate the posterior distribution $p(\theta|\mathbf{x} = \mathbf{x}_0)$. We show that it efficiently learns the parameters of the proposed release model.

## 3   Linear Non-Linear Release Model

Our model consists of two main parts (Fig. 1): a linear-nonlinear (LN) stage models the excitatory drive to the BC and a release (R) stage, models the vesicle pools as dependent random variables (see Appendix A for pseudocode). Therefore, we refer to the model as LNR-model.

### 3.1   Linear-Nonlinear stage

The first stage of the LNR model is a standard LN model, in which a light stimulus $l(t)$ is convolved with a receptive field $w_\gamma$ to yield the surrogate calcium concentration $ca(t)$ in the synaptic terminal which is then passed through a static nonlinearity:

$$ca(t) = \int_{\tau=0}^{T} l(t-\tau) w_\gamma(\tau) d\tau \ .$$

We assume $w_\gamma$ to be a biphasic kernel in order to model the signal processing performed in the photoreceptor and the BC [16, 25] (Figure 1A, B). A single parameter $\gamma$ was used to stretch/compress the kernel on the time axis to estimate the receptive field (see Appendix C). An approach to allow for more flexibility (e.g. using basis function [2]) could in principle be used as well. However, this would lead to a higher dimensional parameter space, making inference less efficient. We used a sigmoidal non-linearity to convert the calcium signal to the release probability:

$$p_{d_t}(t) = 1/\big(1 + \exp(-k(ca_t - h))\big), \tag{1}$$

where the parameters for the slope $k$ and the half activation $h$ are inferred from the data. We add a small positive offset to the non-linearity and renormalized it to allow for spontaneous release.

## 3.2 Release stage

The second stage of the LNR model consists of a model for the synaptic dynamics based on the structure of the BC ribbon: we define variables representing the number of vesicles present in each pool of the ribbon and the number of vesicles moving between pools per timestep (see Table 1). We use capital letters to define the number of vesicles in a specific pool, and lowercase letters to indicate the moving vesicles. At each time step, vesicles are first released from the dock, then new vesicles are moved from the ribbon to the dock, and finally the ribbon is refilled from the cytoplasm. For simplicity, we assume that only the vesicle release probability is modulated by the excitatory drive; the docking probabilities and rates of movement to the ribbon are constant over time.

**Vesicle Release**     To model the correlated release of docked vesicles, we use a beta binomial distribution. This is a binomial distribution for which the parameter $p$ is itself a random variable, leading to correlated events [26]. The release probability $p_{d_t}$ is assumed to be the output of the LN stage according to equation 1. To achieve a correlation $\rho$ for the released vesicles and a release probability of $p_{d_t}$ the parameters for the beta binomial distribution are:

$$\alpha_t = p_{d_t} \cdot \left(\frac{1}{\rho} - 1\right) \quad \text{and} \quad \beta_t = \alpha_t \cdot \left(\frac{1}{p_{d_t}} - 1\right), \quad \text{if } p_{d_t} \neq 0.$$

Thus, in each time step, we first draw the parameter $\tilde{p}_t$ for the binomial distribution according to a beta distribution: $\tilde{p}_t \sim Beta(\alpha_t, \beta_t)$ and then sample the number of released vesicles $d_t$ from a binomial distribution with parameters $n = D_{t-1}$ (the numbers of vesicles at the dock) and $\tilde{p}_t$:

$$p(d_t | D_{t-1}) = \begin{cases} 0 & \text{if } p_{d_t} = 0, \\ \binom{D_{t-1}}{d_t} \tilde{p}_t^{d_t} (1 - \tilde{p}_t)^{(D_{t-1} - d_t)} & \text{otherwise.} \end{cases}$$

**Movement to the dock**     We assume that $r_t$ vesicles located at the ribbon move to the dock in each time step. Because there is a maximum number of vesicles $D_{max}$ that can be docked, such that $r_t + D_{t-1} \leq D_{max}$, we use a restricted binomial distribution to model stochastic vesicle docking:

$$p(r_t | R_{t-1}, D_t) = \begin{cases} \binom{R_{t-1}}{r_t} p_r^{r_t} (1 - p_r)^{(R_{t-1} - r_t)} & \text{if } r_t < D_{max} - D_t \\ \sum_{r_t \geq D_{max} - D_t} \binom{R_{t-1}}{r_t} p_r^{r_t} (1 - p_r)^{(R_{t-1} - r_t)} & \text{if } r_t = D_{max} - D_t \\ 0 & \text{otherwise.} \end{cases}$$

The first case is the standard binomial distribution with appropriate parameters, the second case models the assumption that moving more vesicles to the dock than its maximum capacity simply fills the dock and assures that the probabilities over all possible events sum up to one.

**Movement to the ribbon**     We assume a large number of vesicles available in the cytoplasm (which is not explicitly modeled), such that the number of vesicles $c_t$ moving from the cytoplasm to the ribbon follows a Poisson distribution, again respecting the maximal ribbon capacity $R_{max}$:

$$p(c_t | R_t) = \begin{cases} e^{-\lambda} \frac{\lambda^{c_t}}{c_t!} & \text{if } c_t < R_{max} - R_t \\ \sum_{c_t \geq R_{max} - R_t} e^{-\lambda} \frac{\lambda^{c_t}}{c_t!} & \text{if } c_t = R_{max} - R_t \\ 0 & \text{otherwise.} \end{cases}$$

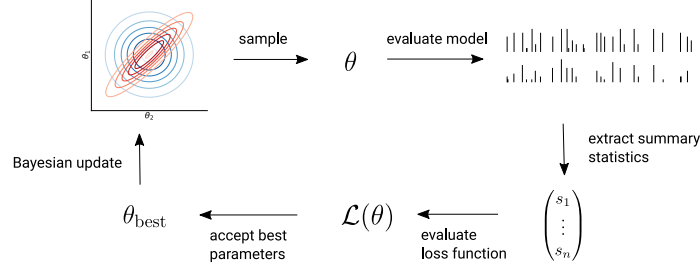

Figure 2: **Overview of the inference method.** In each round samples are drawn from the (proposal) prior (blue), the model is evaluated and the response is mapped to its summary statistic. From this, the loss per parameter $\theta$ is calculated, the best samples are accepted and used to update the (proposal) prior via Bayesian updating rules, yielding a posterior (red), which is the proposal prior for the next round.

# 4   Bayesian Inference of Model Parameters

In the previous section, we constructed a fully stochastic model of vesicle release from BCs, including an explicit mechanistic model of the ribbon synapse, reflecting the underlying biological structures. The maximal capacity of the dock $D_{max}$ was set based on the measured data to the largest quantal event observed in the functional recording ($D_{max} \approx 7 - 8$). $R_{max}$, the maximal capacity of the ribbon, was set to an estimate on the maximal number of vesicles at the ribbon in goldfish rod bipolar cells [27, 28], but decreased to reflect the smaller size of cone BCs in zebrafish larva [29] ($R_{max} \approx 50$).

Next, we developed an ABC framework for likelihood free inference to infer the remaining model parameters (Table 1) from functional two-photon recordings. Our method uses parametric prior distributions which are updated in multiple rounds via Bayesian updating rules to provide a unimodal approximation to the posterior (Figure 2). Briefly, in each round we first draw a parameter vector $\theta$ from the (proposal) prior and evaluate several runs of the model $\hat{d}_i$ for each sampled parameter vector. Due to the stochasticity of the model, each evaluation returns a different trace, for which a summary statistic is calculated. This summary statistic reduces the dimensionality of the simulated trace to a low dimensional vector. Based on this the loss function $\mathcal{L}(\theta)$ is calculated by comparing it to the summary statistic of the observed data. The best parameters are used to calculate a posterior, which is then used as a proposal prior in the next round (Fig. 2, pseudocode in Appendix E).

## 4.1   Prior distributions and inference

As priors, we used normal distributions for all parameters except for $\lambda_c$ (Table 1), where we used a gamma distribution (the conjugate prior to the Poisson distribution). Some parameters were bounded e.g. to the interval $[0, 1]$ and their distributions renormalized to effectively truncate the priors.

In each inference round, we used Bayesian updating rules to calculate the posterior distribution [30, 31] based on the $j$ best parameters $\{\theta\}$. For example, in round $n+1$, we updated the hyperparameters for the multivariate normal distribution of the NL parameters, $k$ and $h$, as:

$$\mu_{n+1} = \frac{\kappa_n}{\kappa_n + j}\mu_n + \frac{j}{\kappa_n + j}\bar{\theta}$$

$$\Lambda_{n+1} = \Lambda_n + S + \frac{\kappa_n j}{\kappa_n + j}(\bar{\theta} - \mu_n)(\bar{\theta} - \mu_n)^T \quad,$$

where $\bar{\theta}$ is the mean over the best parameters and $S = \sum_{i=1}^{j}(\theta_i - \bar{\theta})(\theta_i - \bar{\theta})^T$ the (unnormalized) covariance of these parameters. The mean is thus updated as a weighted average of the prior mean and the mean of the best parameters, with weights specified by $\kappa$, which is updated as $\kappa_{n+1} = \kappa_n + j$. The posterior degrees of freedom $\nu_{n+1}$, which is used to sample the covariance matrix $\Sigma$, is the prior degrees of freedom plus the updating sample size: $\nu_{n+1} = \nu_n + j$. With these updates we end up in a two step sampling procedure: first we draw the covariance $\Sigma_{(n+1)_i}$ for each sample $i$ of round $n+1$ from the inverse-Wishart distribution Inv-Wishart($\Lambda_{n+1}^{-1}, \nu_{n+1}$), and then we draw the samples from the normal distribution $\mathcal{N}(\mu_{n+1}, \Sigma_{(n+1)_i})$.

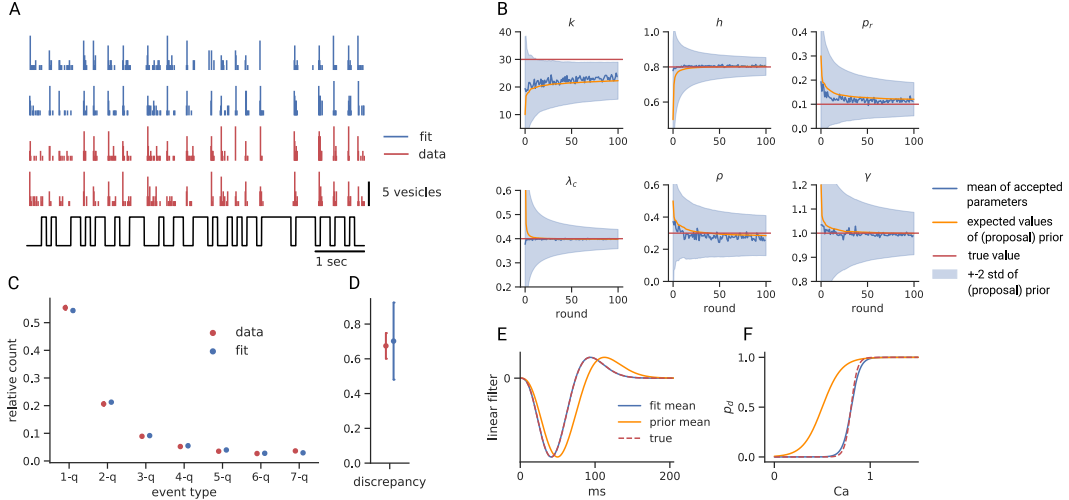

Figure 3: **Results for synthetic data. A.** Simulated traces for the synthetic data and simulations with the recovered, fitted parameters in response to a binary light stimulus. **B.** The time course of the mean and standard deviation of the different one dimensional marginal distributions over several rounds. Notice the asymmetric distribution for $\lambda_c$. See Appendix Fig. 8 for the two dimensional marginals. **C.** Relative count for the different event types, error bars indicating $\pm$ std. **D.** Discrepancy of the data and fitted traces. The discrepancy is defined as the difference between the weighted summary statistics of a single data trace and the remaining data ("leave-one-out-procedure") and accordingly the difference between the weighted summary statistics of a single fitted trace and the data. Error bars indicating $\pm$ std. **E.** The kernel of the linear stage. **F.** The non-linearity. Although its parameter $k$ is not matched perfectly in (B), there is almost no difference between the fitted and the true non-linearities.

The parameters for the univariate normal distributions as well as for the $\Gamma$-distribution are similarly updated in a Bayesian manner (see Appendix D). The number of drawn and accepted parameters was constant ($20 \cdot 10^3$ and 10) except for the first round, where the number of drawn parameters was doubled.

## 4.2 Summary statistics and loss function

As a summary statistic, on which the discrepancy between different traces is defined, we used (1) the histogram over the number of vesicles released in each event and (2) the Euclidean distance between the simulated and measured response trace, convolved with a Gaussian kernel (width: 100 ms, inspired by [32]). The former proved especially useful in early rounds of inference. As experiments typically consist of multiple repetitions of the same stimulus, we first calculated the summary statistics $s(d_i)$ for the individual traces $d_i$, normalized each entry by the summary statistic of the data traces and scaled it for its importance. This linear transformation is summarized in a diagonal weight matrix $W$. We used the average euclidean distance of these weighted summary statistics as the loss function $\mathcal{L}(\theta)$ (see also Appendix E and F). For $n$ data traces $d_i$ and a batchsize of $m$ simulations $\hat{d}_j$ per parameter $\theta$, this yields:

$$\mathcal{L}(\theta) = \frac{1}{nm} \sum_{i,j} ||Ws(d_i) - Ws(\hat{d}_j)||_2 \ .$$

The (weighted) summary statistics can also be used to calculate the variability of the data and compare it to the summary statistics of the simulated data, giving us an estimate of the discrepancy between the different traces (e.g. Fig. 3C, Fig. 5C).

## 4.3 Runtime and complexity

The runtime of the presented ABC method is dominated by the forward simulations of the model, with a complexity $\mathcal{O}(n)$ if $n$ is the number of drawn samples. This complexity is similar to SNPE-B

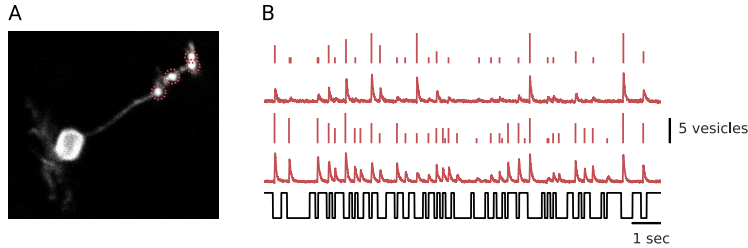

Figure 4: **Two-photon imaging of *in vivo* zebrafish BCs allows for counting glutamatergic vesicles. A.** Image of a zebrafish BC expressing the Superfolder-iGluSnFR transgene. Dashed circles indicate active zones where glutamate is released. **B.** Experimental glutamate release traces as $\Delta F/F$ of one OFF BC in two trials and extracted events in response to a binary light stimulus. Notice the high inter-trial variability.

[6], which in addition requires training of a mixture density network, while we resort to analytic updating formula. Although for expensive simulations, either strategy is often only a small fraction of the total run time, our method should be advantageous if the simulation is fast and the posterior unimodal. This direct estimation of the posterior stands in contrast to SNL [9] or BOLFI [8] where the inference of the posterior involves a second sampling step via MCMC which can be slow. In addition, BOLFI [8] uses a Gaussian process with complexity $\mathcal{O}(n^3)$ in the vanilla version to approximate the likelihood, which can be prohibitively slow.

## 5   Results

### 5.1   Model inference on synthetic data

Next, we tested whether we can successfully infer the parameters of the mechanistic model with the procedure outlined above. For that, we chose a realistic parameter setting and used the model to generate data. As the sample size per cell is severely limited in experimental data, we generated only four traces of 140 seconds each (Fig. 3A). The light stimulus was the same binary noise stimulus that we used for the experimental recordings.

The inference procedure proved very efficient: most parameters converged quickly to their true values with reasonable uncertainty (Fig. 3B). Only the slope parameter $k$ of the non-linearity is underestimated, likely because of the "non-linear" effect of $k$ on the slope of the non-linearity and the smaller prior mean. The method sets $k$ to a value where a further increase woult not change significantly the output of the model (see also Fig. 3F). After inference, it is difficult to differentiate between the true and the fitted traces and the histogram over the number of vesicles released in each time step can be fitted well (Fig. 3C). Indeed, simulations from the model where as similar to the data as different data trials to one another (Fig. 3D). Our procedure identified the time scale of the linear receptive field as well as the non-linearity effectively (Fig. 3E and F). We validated the efficacy of our method also for other choices of parameters (not shown).

### 5.2   Model inference on BC recordings from zebrafish retina

We acquired two-photon imaging measurements of the glutamate release from BC axon terminals in the zebrafish retina ($n = 2$ cells, see Fig. 4). Briefly, linescans (1 x 120 pixels) were recorded at 1 kHz across the terminal of a BC expressing the glutamate reporter Superfolder-iGluSnFr, while showing a 140 second light stimulus (discrete Gaussian or binary noise) with a frame rate of 10 Hz. For each recording, a region of interest (ROI) was defined and the time series extracted as the weighted average of the spatial profile. Baseline drift was corrected, the traces were converted to $\Delta F/F$ and deconvolution was done with the kinetics of the reporter. Release events were identified as local maxima above a user-defined threshold in the imaging trace. The number of vesicles in each release event was estimated using a mixture of Gaussian model. For more details see [33].

Figure 5A shows the LNR model fitted to four recordings from one OFF BC (total duration of the recordings: 560 sec). We find that model parameters both for the LN-stage as well as the release stage of the model can be inferred efficiently. Posteriors converged quickly (Fig. 5B and Appendix Fig.

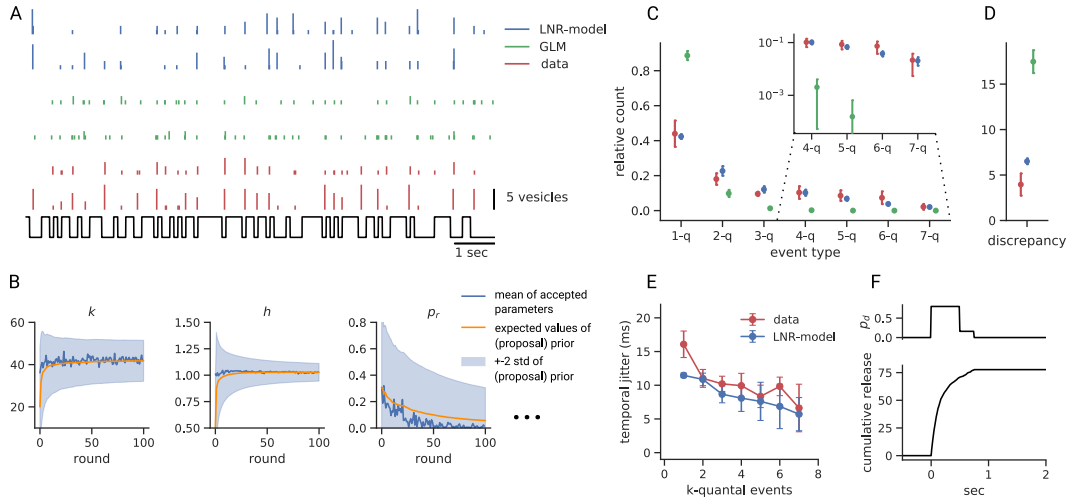

Figure 5: **Results for experimental data. A.** Two experimental data traces and simulated traces with corresponding fitted parameters as well as two predictions from the GLM in response to a binary noise stimulus. **B.** Some parameters are more restricted than others by the model. (See Appendix Section I for all parameters, the two dimensional marginals, and the corresponding kernel and non-linearity) **C.** Relative count for the different event types for the data and the models, inset on log-scale (mean±std). **D.** Discrepancy as in Fig. 3C. **E.** Temporal jitter of different event types in response to a binary light stimulus (mean±std, see Appendix Fig. 7 for the Gaussian noise stimulus). **F.** Cumulative release in response to a "calcium step" (see Appendix Fig. 11 for a comparison to experimental data).

9). Interestingly, parameters such as the ribbon-to-dock-transition rate $p_r$, which model not directly observable properties of the system, also had larger uncertainty estimates. Similar to the synthetic data, the histogram over the number of vesicles released in each event was matched well overall (Fig. 5C). In contrast to the synthetic data, the discrepancy among data traces was a bit smaller than the discrepancy between the model fit and the data traces (Fig. 5D). This is likely due to the fact that some events were missed and the data contained more large amplitude events than predicted by the model (Fig. 5A, C).

We finally tested whether the simple model captured two known properties of release events: the temporal precision of events and the maximal release rates of the system. Interestingly, events with many released vesicles were temporally more precise for both the data and the fitted model (Fig. 5E, F). As no summary statistic explicitly measured the temporal precision of different release event types at this resolution, this can be seen as evidence that our model captures crucial aspects of processing in BCs. Additionally, when comparing release rates with those recorded from electrically stimulated cells, we find the shape of cumulative vesicle release matches well with previously published results (Fig. 5F and Appendix J). This indicates that the model also extrapolates well to new stimuli.

## 5.3 Comparison to a GLM

We compared the prediction performance of the LNR model to a generalized linear model (GLM) [2], a commonly used model in neural system identification. Besides the stimulus term, it includes a self-feedback term and assumes Poisson noise (for details see Appendix K). In contrast to the LNR model, the GLM was not able to capture the MVR that is apparent in the data: The GLM did not predict events with more than five vesicles at all, and already events with more than two vesicles were rare (Fig. 5A,C). This results in much larger discrepancies overall compared to the LNR model (Fig. 5D). The weights of the linear part for the release history captured the suppression of additional release after a release event partly, but could not model the full dynamics (Appendix Fig. 12C and Fig. 5A). This shows that supplementing systems identification models with biophysical components can dramatically improve prediction accuracy, not only lead to more interpretable models.

# 6 Discussion

Here we developed a Bayesian inference framework for the LNR model, a probabilistic model of vesicle release from BCs in the retina, which combines a systems identification approach with a mechanistic, biophysically inspired component. In contrast to purely statistical models, the parameters of the LNR model are readily interpretable in terms of properties of the ribbon synapse. Specifically, we show that its parameters can be fitted efficiently on synthetic data and two-photon imaging measurements. The latter is remarkable, as mechanistic models often require highly specialized experiments to determine individual parameters. In this proof-of-principle study, we show that the parameters of the LNR model can be simply inferred from the functional measurements, opening possibilities for inferring mechanistic models from large-scale imaging experiments e.g. for comparison across cell types.

We found that the data overall was able to constrain the parameters very well, for both the LN-stage and the release-stage of the LNR model. Parameters that referred to parts of the model that were not directly observed in our measurements (such as the transition probability from the ribbon to the dock $p_r$) were fitted with somewhat higher uncertainty, indicating that a larger range of parameter values was compatible with the measurements. In addition, the LNR model captured MVR (the inferred correlation between vesicles is $\rho \approx 0.35$), despite the inherent variability at the level of the single synapse. In addition, the LNR model captured trends in temporal precision within MVR events as well as release rates to non-phyisiological stimuli such as electrical stimulation - neither of which were used during inference.

Our proposed framework for Bayesian inference in the LNR model is comparable to recent likelihood-free inference methods (e.g. [6, 11]). In contrast to those, we do not use a mixture density network (MDN) to approximate the posterior distribution, but rather parametric distributions and analytic Bayesian updating rules. In practice, MDNs can lead to unstable behavior for very small or large weights and sometimes have non-optimal extrapolation properties (but see [34]). Due to its simplicity, our method is less susceptible to such problems, but provides only a unimodal approximation of the posterior $p(\theta|\mathbf{x} = \mathbf{x}_0)$. For the LNR model, we rarely observe multimodality in the posterior, so our methods yield a good and very fast approximation to the true posterior.

We combined a biophysically inspired mechanistic model with an efficient likelihood free inference method. This eases the development of more accurate and interpretable models, without the necessity of having closed-form likelihoods. At the same time, we could show that our model is able to capture the variablity inherent to the neural system we studied and that taking biophysical constraints into account can even dramatically improve prediction accuracy of system identification models compared to standard systems identification models. Taken together, the presented methods will allow for further investigations into more complex systems, gaining mechanistic insight into how neurons cope with noise.

**Acknowledgments**

We thank Sofie-Helene Seibel for help in the experiments and genetics for this project and for the BC image in Fig. 4A, Christian Behrens for providing the PR/BC schema in Fig. 1B, Jan Lause for his detailed feedback on the manuscript. The study was funded by the German Ministry of Education and Research (BMBF, 01GQ1601, 01IS18052C and 01IS18039A) and the German Research Foundation (BE5601/4-1, EXC 2064, project number 390727645). In addition, this project has received funding from the European Union's Horizon 2020 research and innovation programme under the Marie Skłodowska-Curie grant agreement No 674901 and the Wellcome Trust (Investigator Award 102905/Z/13/Z).

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
