[Supplementary Material · Schroeder_and_James_appendix.pdf]

# Appendix

## A    Pseudocode of the linear-non-linear release model

---

**Algorithm 1:** Pseudocode for the LNR-model

---

Calculate output of LN-block: $p_{d_t} = \text{NL}(w_\gamma(t) * l(t))$;
Initialize pools $D_0, R_0$;
**for** $t \in (1, \text{timesteps})$ **do**
>  Draw $d_t \sim$ beta Binomial$(D_{t-1}, \rho, p_{d_t})$;
>  $D_t = D_{t-1} - d_t$;
>  Draw $r_t \sim$ restricted binomial$(R_{t-1}, p_r | D_t)$;
>  $D_t = D_t + r_t$;
>  $R_t = R_{t-1} - r_t$;
>  Draw $c_t \sim$ restricted Poisson$(\lambda_c | R_t)$;
>  $R_t = R_t + c_t$

**end**

---

## B    Experimental Methods

The data were collected and analyzed as described in [33] in accordance with the Animal Act 1986 and the UK Home Office guidelines and with the approval of the the University of Sussex local ethical committee. RibeyeA:Gal4 transgenic zebrafish were crossed with 10 x UAS:Superfolder iGluSnFR fish [35]. The resulting larva were imaged at 5-7 days post fertilization in vivo after embedding in 5 percent low-melting agarose. Imaging was done with a Scientifica microscope using a Coherent laser tuned to 915 nm. All stimuli were delivered full-field with an LED (lmax = 590 nm, Thorlabs), filtered through a 590/10 nm BP filter (Thorlabs), and output through a light-guide, resulting in a mean intensity of approximately 316 nW/mm$^2$. Two white noise stimuli were used for the experiments: a binary stimulus constructed using an m-sequence protocol, and a discrete Gaussian stimulus, both with a frame rate of 10 Hz. Two cells recorded from two eyes of a single fish were used for the experiment.

## C    Linear kernel

Figure 6: Example kernels for 100 samples of compress/stretch parameter $\gamma$, drawn from the prior distribution ($\gamma \sim \mathcal{N}(1, 0.2)$), and the corresponding kernels for the mean (red) and $\pm 2$ standard deviations (dashed red). The remaining parameters were held constant: $\tau_r = 0.05$, $\tau_d = 0.05$, $\tau_{\text{phase}} = 100$, $\phi = -\pi/7$.

The linear part of the model is the convolution of a kernel with the (light) stimulus $l(t)$:

$$ca(t) = \int_{\tau=0}^{T} l(t-\tau)w_\gamma(\tau)d\tau \ .$$

We used a typical kernel for photoreceptors from previous work [16, 25], which depends on a rise and decay constant ($\tau_r$ and $\tau_d$) and additional phase parameters ($\phi$ and $\tau_{\text{phase}}$). To get different time scales, we included a new parameter $\gamma$ that stretches or compresses the kernel in time and was inferred from the data. Including $\gamma$, the equation of the kernel becomes:

$$w(t, \gamma) = \frac{-\left(\frac{t}{\gamma \tau_r}\right)^3}{1 + \frac{t}{\gamma \tau_r}} \cdot \exp\left(-\left(\frac{t}{\gamma \tau_d}\right)^2\right) \cdot \cos\left(\frac{2\pi t}{\gamma \phi} + \tau_{\text{phase}}\right) \ .$$

Figure 6 shows example kernels for different draws of $\gamma$.

Note, that the learned filter in the LNR-model is different from the filters recovered with e.g. the event triggered average, as the release dynamics are taken into account in its estimation.

## D Detailed updating rules

We follow the standard Bayesian updating rules to calculate the posterior [30, page 73].

**Normal distributions** As already stated, we use a two dimensional multivariate normal distribution with unknown mean and covariance as distribution for the non-linearity parameters $k$ and $h$. For the remaining parameters, we take independent one dimensional normal distributions with unknown mean and variance (except for $\lambda_c$), such that the updating rules are basically the same: in round $n+1$, we take the $j$ best parameters $\{\theta\}$ and update as follows:

$$\mu_{n+1} = \frac{\kappa_n}{\kappa_n + j}\mu_n + \frac{j}{\kappa_n + j}\bar{\theta}$$
$$\kappa_{n+1} = \kappa_n + j$$
$$\nu_{n+1} = \nu_n + j$$
$$\Lambda_{n+1} = \Lambda_n + S + \frac{\kappa_n j}{\kappa_n + j}(\bar{\theta} - \mu_n)(\bar{\theta} - \mu_n)^T$$

where $\bar{\theta}$ is the mean of the best parameters and $S = \sum_{i=1}^{j}(\theta_i - \bar{\theta})(\theta_i - \bar{\theta})^T$ the (unnormalized) estimation for the covariance of the best parameters.

To sample from the posterior distribution, we first draw $\Sigma$ from the inverse-Wishart distribution $\Sigma \sim \text{Inv-Wishart}(\Lambda_{n+1}^{-1}, \nu_{n+1})$, and then we draw samples from $\mathcal{N}(\mu_{n+1}, \Sigma)$.

For the one dimensional normal distributions we follow the same updating rules, except that the variance $\sigma_{n+1}^2$ is updated as following:

$$\sigma_{n+1}^2 = \frac{1}{\nu_n}\left(\nu_n \sigma_n^2 + S + \frac{\kappa_n j}{\kappa_n + j}(\bar{\theta} - \mu_n)^2\right)$$

To sample from the posterior we first draw $\sigma^2 \sim \text{Inv-}\chi^2(\nu_{n+1}, \sigma_{n+1}^2)$ and then draw samples from $\mathcal{N}(\mu_{n+1}, \sigma^2)$.

**Gamma distribution** The parameter $\lambda_c$ for the Poisson distribution has a $\Gamma$-distribution as conjugate prior, which is as well updated in a Bayesian way. We update the hyperparameters $k$ and $\theta$ of the $\Gamma$ distribution for the $j$ best parameters $\{\lambda_i\}_{i=1,\dots,j}$ as:

$$k_{n+1} = k_n + \sum_{i=1}^{j} \lambda_i$$
$$\theta_{n+1} = \frac{\theta_n}{(1 + j \cdot \theta_n)} \ .$$

# E Pseudocode of the ABC method

**Algorithm 2:** Parametric approximation of the posterior via Bayesian updating

---

**Input:** Simulator $p(x|\theta)$; data traces $\{x_{0_i}\}_{i \in I}$; Summary statistics $s(x)$ with weights $W$; prior
distribution $p(\theta)$; number of rounds $R$; simulations per round $N$; batchsize $B$, number of
accepted parameters $n_{\text{best}}$.

$\tilde{p}_1(\theta) := p(\theta)$ ;
**for** $r \in (1, R)$ **do**
    **for** $n \in (1, N)$ **do**
        Draw $\theta_{r,n} \sim \tilde{p}_r(\theta)$ ;
        **for** $b \in (1, B)$ **do**
            Simulate $\hat{x}_{r,n,b} \sim p(x|\theta_{r,n})$ ;
        **end**
        $\mathcal{L}(\theta_{r,n}) := \frac{1}{IB} \sum_{i,b} ||Ws(x_{0_i}) - Ws(\hat{x}_{r,n,b})||_2$ ;       // compute loss
    **end**
    $\theta_{\text{sorted}} := \text{sort}(\{\theta_{r,n}\}_{n \in N})$ ;       // sort by loss function
    $\theta_{\text{best}} := \theta_{\text{sorted}}[0 : n_{\text{best}}]$ ;       // accept best samples
    $\tilde{p}_{r+1}(\theta) := \tilde{p}_r(\theta|x_0, \theta_{\text{best}})$ ;       // Bayesian updating of (proposal) prior
**end**
**Return:** $\tilde{p}_{R+1}(\theta)$

---

# F Details for the inference method

For the presented data we run 100 rounds with $20 \cdot 10^3$ samples per round (except for the first round, where we used $40 \cdot 10^3$ samples) and a batchsize of 4 simulated data traces per sampled parameter. We accepted the 10 best parameters per round (the ones producing the lowest loss) and used these to update our (proposal) prior.

For the prior distributions we used the following hyperparameters:

For the multivariate distribution for the non-linearity we used

$$\mu_0 = (20, 0.5)$$
$$\kappa_0 = 4$$
$$\nu_0 = 4$$
$$\Lambda_0 = \text{Id}_2 \cdot \begin{pmatrix} 400 \\ 0.1 \end{pmatrix} \quad ,$$

and truncated the distribution to the domain $(0, 50) \times (-2, 3)$.

For the univariate distribution for $p_r$ $(\rho, \gamma)$ we used

$$\mu_0 = 0.3 \ (0.5, \ 1)$$
$$\kappa_0 = 3$$
$$\nu_0 = 3$$
$$\sigma_0^2 = 0.05 \ (0.05, \ 0.2) \quad ,$$

and truncated the distribution to the interval $(0, 1)$ $((0, 1), \ (0.05, 2))$.

For the Gamma distribution $\Gamma(k, \theta)$ we specified $k_0 = 2$, $\theta_0 = 0.25$ ($\mathbf{E}[\lambda] = k\theta$) and bounds $(0, 1)$ as additional prior information.

As described in the main text we took the histogram over the different event types as summary statistics as well as the euclidean distance of the traces convolved with a Gaussian kernel $g$. As additional entry we took the overall released vesicles. As scalings for the summary statistics we took $\text{Id}_8 \cdot (5, 5, 5, 5, 2, 2, 4, 2)^t$ in the order ($|d_i * g - \hat{d}_j * g|_2$, total release, $\sum \mathbf{1}_{\hat{d}_j=1}, ..., \sum \mathbf{1}_{\hat{d}_j=6}$), where $d_i$ are the data traces and $\hat{d}_j$ is the simulated data. The weight matrix $W$ is the product of the normalizing constants, which is the mean summary statistic of the data traces, with these scalings. The weights for the summary statistics were chosen to make some features more important, but our

experiments suggest that inference results were largely insensitive to the exact choice. While we chose the weights heuristically, in principle, cross-validation could be used for a more systematic procedure.

## G    Temporal jitter

Figure 7: **Temporal jitter for the response to the Gaussian noise stimulus.** Same as Fig. 5E, but here for a Gaussian noise stimulus. As in previous studies [33], the lower contrast stimulation of the Gaussian noise stimulus results in higher temporal jitter when compared to the binary noise stimulus. This is well captured by the model. The exact numbers differ slightly to previous studies where a sinusoidal stimulus was used instead of binary and binned Gaussian noise.

In order to define the temporal precision of release events, we begin by computing the vector strength for each quantal event type, $VS_q$:

$$VS_q = \frac{1}{N_q} \sqrt{ \left( \sum_{i=1}^{N_q} \cos\left( \frac{2\pi t_{q_i}}{T} \right) \right)^2 + \left( \sum_{i=1}^{N_q} \sin\left( \frac{2\pi t_{q_i}}{T} \right) \right)^2 } \ ,$$

where $N_q$ is the number of q-quantal events, $t_{q_i}$ is the phase of the $i$-th $q$-quantal event with respect to the stimulus, and $T$ is the stimulus period. This metric takes values from zero to one, with zero representing complete independence between event times and stimulus phase, while a value of one indicates perfect phase-locking with all events occurring at exactly the same phase.

Temporal precision is more commonly expressed as temporal jitter, the standard deviation in milliseconds of the phase of the event times, as this yields a more intuitive interpretation. Vector strength can be mapped to temporal jitter by a one-to-one function:

$$TJ_q = \frac{\sqrt{2(1 - VS_q)}}{2\pi f} \ ,$$

where $f$ is the stimulus frequency.

# H    Results for the synthetic data

Figure 8: **Approximated marginals for the synthetic data.** One and two dimensional marginals of the posterior distribution for all parameters inferred from synthetic data in response to a binary noise stimulus.

# I Results for the experimental data

Figure 9: **Marginals of the posterior for the experimental data (binary noise stimulus). A.** The time course of the mean and standard deviation of the one dimensional marginal distributions. **B.** One and two dimensional marginals of the posterior.

Figure 10: **LN portion of the model for experimental data. A.** Linear kernel according to the prior and the posterior for the response to the binary noise stimulus. **B.** Non-linearity according to the prior and the posterior for the response to the binary noise stimulus.

## J    Comparison to experimentally measured release rates

Figure 11: **Inferred parameters reproduce the overall shape of vesicle release, as compared with previously published results. A.** Cumulative release rates for the model fit to experimental data bypassing the LN block (same as Fig. 5F). **B.** Estimates of the cumulative release from a goldfish rod bipolar cell in response to electric stimulation (taken from [36]).

Previous work has estimated maximum release rates from ribbon synapses by electrically stimulating the cell and estimating the cumulative number of vesicles released [36]. In order to compare these results with those estimated by fits of our model on experimental data, we bypassed the LN portion of our model, instead setting the release probability parameter $p_d$ to mimic electrical stimulation. $p_d$ was set to 0.8 during the period to simulate electrical stimulation, and 0.2 during the 250 ms period after the stimulation to mimic lingering increases in internal calcium (based upon experimental recordings [36]). All other parameters were set as fit with experimental data.

While the cumulative release between previous reports and our model vary by a constant scaling factor, this is likely due to the stimuli used to fit the model to experimental data. The maximal release rates estimated using physiological stimuli differ from those estimated using direct electrical stimulation.

## K    Details for the GLM

Figure 12: **Baseline GLM A.** Release probability as predicted by the Poisson stage of the GLM and released vesicles in response to a binary stimulus for one example simulation. Notice the tiny refractory period after a release event in the release probability trace. **B.** Weights for the linear part of the GLM for the stimulus (mean $\pm$ std. over four trials). **C.** Weights for the linear part of the GLM for the release history with small suppression term for the first $\sim 30$ ms (mean $\pm$ std. over four trials).

We used a GLM as a baseline model, assuming Poisson noise and a logarithmic link function. The model was trained to predict the release from the last 100 ms of the stimulus as well as 100 ms of the release history. The fitted weights are shown in Fig. 12B,C. We used the `glmnet` function implemented by the python package `glmnet_python` (https://github.com/bbalasub1/glmnet_python.git) with hyperparameters `alpha=0` representing pure $L_2$-regularization and a regularization parameter `s=0.02`.