[Reviews · NeurIPS 2019]

Reviewer 1



[Edit: I read all reviews and the author responses, and still think this is a great work. The author responses answered my questions as well as points raised by other reviewers, providing additional clarification.] This paper formulates a fully probabilistic model of the vesicle-release dynamics at the sub-cellular biophysical level in the ribbon synapse. The paper then develops a likelihood-free inference method, tests it on a synthetic dataset, and finally infers the parameters of vesicle release in the ribbon synapse from real data. Originality: The paper presents a novel combination of biophysical modeling of ribbon synapse and a likelihood-free inference of the parameters. To my knowledge, the fully stochastic modeling of the vesicle-release dynamics is itself new. The likelihood-free inference is based on the recent developments of related methods, but includes problem-specific constructions such as the use of certain summary statistics in the loss function. Quality: The formulation is technically strong; each step is well supported, with the derivations provided in the supplementary material. Modeling assumptions are clearly stated. The overall claim of the paper (effectiveness of the inference) is first confirmed on a synthetic dataset, then is further supported by the observation of known trends in the real-data application. Clarity: This paper is superbly written. Materials are logically organized. There is a good balance between the technical content, and appropriate comments/interpretations to keep the reader on track and provide relevant insights. Significance: The work is significant in two important ways. First, it develops a solid, analytic method for approximate Bayesian inference, which may be further extended to other systems in future research. Although the success of this particular problem does not guarantee another successful application of the method to a more complicated problem, this is certainly an encouraging first work. In addition, I would be interested to know about the running time / complexity of the inference algorithm. Second, the inference result provides a new mechanical understanding of the release dynamics of the ribbon synapse, to a level that was not possible before; the result will serve as a baseline for further biophysical investigations. Minor comments: - Line 102 / the numbers of vesicles moving between pools: It is important to state explicitly that these are the numbers "per unit time", or even better, call these "rates". - Paragraph "Vesicle release": it is not clear how the correlation \rho is defined. - Please number the equations. - Lines 124-126: references for D_max and the (reduced) R_max?

Reviewer 2



This paper introduces a likelihood-free inference algorithm for estimating the parameters governing vesicle release in a ribbon synapse. This is based on a mechanistic model of exocytosis that involves several stages of nonlinear stimulus filtering and synaptic vesicle movement, docking, and release. Bayesian inference in this regime is challenging: due the great number of model components and biophysical constraints, even evaluating the likelihood function can be intractable. The authors address this problem by adopting a likelihood-free (also known as approximate Bayesian computation, ABC) approach, where, by repeatedly sampling from a prior distribution, the forward model is simulated many times and the parameters that well-describe the observed data are retained and used to update the prior, and the process repeats. The modelling work is interesting and to a high standard. The writing is clear and motivates the problem well. This work appears to be a novel application of ABC to the problem of vesicle release modelling, although it's not clear if the approach is actually better than previous methods as no comparisons are performed. Major comments: - The parameter k is often underestimated (Fig 3B and Fig 8) yet seems to have a minimal effect on the resulting nonlinearity. It would be beneficial to mention why the slope of the nonlinearity is difficult to obtain in the main text. - What are the different colours in the density plots in Figure 2? Presumably one is the updated prior. - Should p_d in the equation after line 110 be p_{d_t}? This was confusing on a first read as I was searching for p_d among the p_t, p_{d_t}, p(d_t | D_{t - 1}) etc. - It would be helpful to state something along the lines that the summation in the equation after line 113 is to ensure the probabilities sum to 1, as this was not immediately clear to me. - I'm aware that selection of summary statistics is something of an art in ABC methods, but are the results very sensitive to the scalings (line 377)? The different release counts are scaled differently. Is that to account for the more infrequent release counts? Minor comments: - Line 73: Is it that the likelihood function cannot be evaluated at all, or more that evaluation of the likelihood function is inefficient/intractable? - Figure 4A caption: releaed -> released - Figure 6 caption: hold constant -> held constant - Appendix line 362: Detailes -> Details - Line 376: "for for" -> for Update following rebuttal: The authors have addressed all of my concerns and I maintain my support for accepting this paper.

Reviewer 3



The paper is an original application of likelihood free inference to parameter estimation in a mechanistic model of the ribbon synapse. It is not super advanced on the inference aspects, but well executed and described. My feeling is that the community which would be most excited would be ribbon synapse neuroscientists, and they would be most excited about the results of the parameter estimation rather than the method itself. I'm not sure the neuroscience NeurIPS community at large would get that much out of the paper which is not already covered by existing (and more advanced) likelihood free inference papers such as ref [6] already cited in the paper. A few questions: Is the linear filter really constrained to have a single form with a single stretch parameter? I would have assumed that given the discrete Gaussian or binary noise stimulus, it would be possible to estimate the full filter directly from data. Why is the filter constrained in this way? It appears that the posterior over model parameters inferred via the method is a factorized distribution. What is the reason for leaving out the correlations? I would have assumed that it would be relatively straightforward, given that the distributions are all Normal? Is it because of the truncation applied to some of the distributions? The acceptance criterion is to accept the best j particles. I wonder how well an acceptance criterion based on the loss value would work? Is it possible that the variance of the posterior is over or under estimated by the best j criterion? For instance, if more than j particles are within an acceptable loss value, then we would get an underestimate. And conversely, if the worst particles in the j best particles have unacceptable loss values, then the variance will have been over estimated.

[Author Response · NeurIPS 2019]

We thank the reviewers for their insightful comments regarding our paper. R1 and R2 highlighted the technical quality and clarity of our work and the novelty of the application. All minor comments will be addressed in the revised paper. Here, we briefly reply to selected major points raised by the reviewers (references refer to the main paper):

**Significance of the paper to the NeurIPS community (R3)**
Our work shows that models for system identification, which are widely used in the NeurIPS community, can be supplemented by biophysically inspired components, such as a model of vesicular release at a ribbon synapse, and inference for such models can be performed in a Bayesian manner. Our additional analysis (see point 2 below) shows that our model clearly outperforms GLM-style models for the type of data studied here. On the technical side, we show that for such models which are fast in evaluating, already a simple ABC method can estimate the posterior distribution efficiently and no additional overhead such as training a DNN or GP - like in reference [6] and [8] - is necessary.

**Comparison with previous models (R2)**
To address this point, we performed additional analysis and compared the LNR-model to a GLM with stimulus and self-feedback term and Poisson noise [2]. In contrast to the LNR-model, the GLM was not able to capture the multiple vesicular release with more than three vesicles at a time and showed much larger discrepancies overall ($18.3 \pm 1.8$, mean $\pm$ std compared to $6.5 \pm 0.3$ for the LNR-model). The weights of the linear part for the release history captured the suppression of additional release after a release event partly but could not model the full dynamics. This analysis demonstrates that taking biophysical constraints into account can dramatically improve prediction accuracy of system identification models. We will include an appropriate figure and details in the revised paper.

**Parametrization of the model (R2, R3)**
*Stimulus filter*:
The learned filter in the LNR-model is different from the filters recovered with e.g. the STA, as the release dynamics are taken into account in its estimation. For simplicity, we assumed a stimulus kernel with one parameter only, but a basis function approach [2] to allow for more flexibility could in principle be used as well. However, this would lead to a higher dimensional parameter space, making inference less efficient. Exploring this trade-off further is an interesting direction for future work.
*Slope parameter*:
The slope parameter $k$ of the non-linearity is indeed underestimated, likely because of the "non-linear" effect of $k$ on the slope of the non-linearity. Our method sets $k$ to a value where a further increase would not change significantly the output of the model.
*Summary statistics*:
Indeed, the weights for the summary statistics were chosen to make some features more important, but our experiments suggest that inference results were largely insensitive to the exact choice. While we chose the weights heuristically, in principle, cross-validation could be used for a more systematic procedure.
We will improve our discussion of all three aspects in the revised version of the paper.

**Form of the posterior and acceptance strategy (R3)**
Initial experiments showed almost uncorrelated posterior distributions for most of the parameters. Hence we decided to factorize the distribution in most dimensions and modeled only the non-linearity parameters as a multivariate normal distribution. In general, the described formulas for the two dimensional multivariate distribution would indeed generalize straight forward to higher dimension. However, distributions such as the $\Gamma$-distribution for $\lambda_c$ would then have to be approximated.
Using a fixed acceptance threshold for the loss results in inefficient updates of the proposal prior in early iterations as very few or even no samples are accepted in each round. Using an adaptive threshold might remedy this, but would likewise affect the estimated variance.

**Runtime and Complexity (R1)**
The runtime of the presented ABC method is dominated by the forward simulations of the model, with a complexity $\mathcal{O}(n)$ if $n$ is the number of drawn samples. This complexity is similar to SNPE-B [6], which in addition requires training of a mixture density network, while we resort to analytic updating formula. Although for expensive simulations, either strategy is often only a small fraction of the total run time, our method should be advantageous if the simulation is fast and the posterior unimodal. As already mentioned in the main text, the direct estimation of the posterior stands in contrast to SNL [9] or BOLFI [8] where the inference of the posterior involves a second sampling step via MCMC which can be slow. In addition, BOLFI [8] uses a Gaussian process with complexity $\mathcal{O}(n^3)$ in the vanilla version to approximate the likelihood, which can be prohibitively slow. Additional discussion will be added to the paper.

[Meta-Review · NeurIPS 2019]

This is an interesting paper on a mechanistic model of the ribbon synapse along with an ABC inference approach. Neither component is particularly novel, but the paper is thorough and compelling. The audience will likely be computationally-savvy experimental neuroscientists and those interested in applications of ABC; the former may be harder to find at NeurIPS, though they do exist. I encourage the authors to make the suggested revisions before the camera ready deadline.